# Facile One-Pot Green Synthesis of Magneto-Luminescent Bimetallic Nanocomposites with Potential as Dual Imaging Agent

**DOI:** 10.3390/nano13061027

**Published:** 2023-03-13

**Authors:** Radek Ostruszka, Denisa Půlpánová, Tomáš Pluháček, Ondřej Tomanec, Petr Novák, Daniel Jirák, Karolína Šišková

**Affiliations:** 1Department of Experimental Physics, Faculty of Science, Palacký University Olomouc, 77900 Olomouc, Czech Republic; 2Faculty of Health Studies, Technical University of Liberec, 46117 Liberec, Czech Republic; 3Department of Analytical Chemistry, Faculty of Science, Palacký University Olomouc, 77900 Olomouc, Czech Republic; 4Regional Centre of Advanced Technologies and Materials, Czech Advanced Technology and Research Institute, Palacký University Olomouc, 77900 Olomouc, Czech Republic; 5Radiodiagnostic and Interventional Radiology Department, Institute for Clinical and Experimental Medicine, 14021 Prague, Czech Republic

**Keywords:** nanocomposite material, imaging, gold nanocluster, luminescence material, MRI assessment, SPION, bovine serum albumin

## Abstract

Nanocomposites serving as dual (bimodal) probes have great potential in the field of bio-imaging. Here, we developed a simple one-pot synthesis for the reproducible generation of new luminescent and magnetically active bimetallic nanocomposites. The developed one-pot synthesis was performed in a sequential manner and obeys the principles of green chemistry. Briefly, bovine serum albumin (BSA) was exploited to uptake Au (III) and Fe (II)/Fe (III) ions simultaneously. Then, Au (III) ions were transformed to luminescent Au nanoclusters embedded in BSA (AuNCs-BSA) and majority of Fe ions were bio-embedded into superparamagnetic iron oxide nanoparticles (SPIONs) by the alkalization of the reaction medium. The resulting nanocomposites, AuNCs-BSA-SPIONs, represent a bimodal nanoprobe. Scanning transmission electron microscopy (STEM) imaging visualized nanostructures with sizes in units of nanometres that were arranged into aggregates. Mössbauer spectroscopy gave direct evidence regarding SPION presence. The potential applicability of these bimodal nanoprobes was verified by the measurement of their luminescent features as well as magnetic resonance (MR) imaging and relaxometry. It appears that these magneto-luminescent nanocomposites were able to compete with commercial MRI contrast agents as MR displays the beneficial property of bright luminescence of around 656 nm (fluorescence quantum yield of 6.2 ± 0.2%). The biocompatibility of the AuNCs-BSA-SPIONs nanocomposite has been tested and its long-term stability validated.

## 1. Introduction

Today, nanocomposites that are simultaneously luminescent and magnetically active are the focus of many research groups due to their applications in nanomedicine (for instance, [1,2,3,4,5]). Several approaches can be used to combine luminescent and magnetic features within one nanocomposite: (i) luminescent nanostructures (NSs) (e.g., quantum dots and/or AuNSs) connected with magnetic NSs [6,7,8,9,10,11,12]; (ii) fluorescent (organic) dyes and magnetic NSs [13,14]; (iii) luminescent NSs and magnetic complexes (e.g., containing Gd^3+^) [3,15,16]; and (iv) fluorescent dyes and magnetic complexes [17,18]. Here, we deal with the first approach (luminescent NSs and magnetic NSs) to achieve magneto-luminescent nanocomposites serving as dual (bimodal) probes. 

Typical synthetic strategies for the fabrication of such magneto-luminescent nanocomposites can include: (a) complex multi-step synthesis via a series of sequential synthetic procedures with separately optimized steps (e.g., [6,7]); (b) one-pot method from as-prepared or commercially available structures (e.g., post-synthetic modifications) [13,14,16,17]; and/or (c) a one-pot method without the previous preparation of NSs components (e.g., [15,18]). Here, a straightforward synthesis of the (c) type with a high yield is presented. 

Aside from the complexity of the preparation, the individual syntheses of NSs also differ in the total preparation time, ranging from a few minutes (in the case of microwave-assisted synthesis) [19,20,21] to tens of hours [7,15,18]. Today, simplicity, reproducibility, and green chemistry in NSs preparation are beneficial and highly recommended and are therefore applied in this work. Indeed, we have chosen a protein templated synthesis of luminescent Au nanoclusters based on our previous experience [19,22]. Bovine serum albumin (BSA), a transportation protein which is structurally analogous to human serum albumin, is successfully employed as a matrix for the formation of non-toxic luminescent Au nanoclusters embedded in BSA [19,22]. 

Furthermore, the same protein, BSA, has also been used by other authors in the generation of superparamagnetic iron oxide nanoparticles (SPIONs), which play a special role in the *in vivo* visualization of cells or biological tissues by ^1^H MRI (magnetic resonance imaging) [23,24]. BSA in conjunction with SPIONs are exploited for two reasons: (i) achieving a better *in vivo* biocompatibility (e.g., [25,26,27,28]) and (ii) prolonging the blood circulation lifetime of SPIONs, representing MRI nanoprobes (e.g., [29,30,31,32,33]). Both properties are superior in SPIONs in comparison to, for instance, Gd (III) species, which are exhaustively reported in the literature, even in combination with AuNSs (e.g., [15,16]). Since Gd (III) species are toxic and represent potential risk to environment and human health [34], we instead decided to exploit SPIONs as MRI contrast agents in our nanocomposites. Wang Y. and co-authors [29] generated ultrasmall SPIONs directly in the presence of BSA under alkaline pH, i.e., using a one-step bio-mineralization method. In the works of other authors, BSA created only a part of the modification layers of SPIONs [30,31,32,33,35,36,37]. Nevertheless, none of these SPIONs-BSA nanocomposites manifested fluorescent properties in the visible region of the electromagnetic spectrum. 

In the present study, a one-pot simultaneous bio-mineralization method of gold and iron ions in the presence of BSA under alkaline medium was developed to create new magneto-luminescent probes (further abbreviated as AuBSA-Fe). We demonstrate here that an easy, reproducible, highly efficient synthesis of new functional bimodal probes can be achieved by performing the one-pot sequential preparation procedure. Importantly, in comparison to most of the related literature [12], no abundant chemical agents are necessary and the use of organic solvents was totally avoided by us. Therefore, the synthesis can be regarded as a green one.

Several basic, as well as sophisticated experimental techniques, were exploited for the characterization of our bimodal AuBSA-Fe probes, such as steady-state fluorescence, dynamic light scattering (DLS), UV–Vis absorption measurements, scanning transmission electron microscopy (STEM), energy dispersive spectroscopy (EDS), Mössbauer spectroscopy, inductively coupled plasma mass spectrometry (ICP-MS), relaxation rates determination, and magnetic resonance imaging (MRI). Moreover, cell viability tests were performed by using Alamar blue assay (resazurin) and the long-term stability of AuBSA-Fe nanocomposites was verified by X-ray photoelectron spectroscopy (XPS), among others. 

Our results clearly demonstrate that our AuBSA-Fe probes prepared by a simple one-pot sequential green synthetic procedure are superior to commercial MRI contrast agents owing to their bright luminescence at 656 nm when excited in the visible region (e.g., using 480 nm excitation wavelength).

## 2. Materials and Methods

### 2.1. Chemicals for Syntheses

Bovine serum albumin (BSA; >98%), gold(III) chloride trihydrate (HAuCl_4_·3H_2_O, ≥99.9%), iron(II) chloride tetrahydrate (FeCl_2_·4H_2_O; containing 93.4% of FeCl_2_ and 6.6% of FeOOH according to Mössbauer spectroscopy), iron(III) chloride hexahydrate (FeCl_3_·6H_2_O; ≥99%), and sodium hydroxide (NaOH; ≥98.0%) were purchased from Sigma-Aldrich (Saint Louis, MO, USA) and used as received (without any further purification) for all experiments. Nitric acid (69%, Analpure), hydrochloric acid (36%, Analpure), acid-certified reference materials of the calibration standard solution ASTASOL^®^ of Au, Fe (1000 ± 2 mg·L^−1^), and INT-MIX 1 (10.0 ± 0.1 mg·L^−1^) were purchased from Analytika, Ltd., Prague, Czech Republic, and used only for ICP-MS analyses. Deionized (DI) water prepared by purging Milli-Q purified water (Millipore Corp., Bedford, MA, USA) was used in all experiments.

### 2.2. Chemicals for Alamar Blue Assay

Foetal bovine serum (FBS), L-glutamine, Penicillin-Streptomycin, sodium chloride (NaCl; ≥99.0%), potassium chloride (KCl; ≥99.0%), potassium dihydrogen phosphate (KH_2_PO_4_; ≥99.0%), disodium hydrogen phosphate (Na_2_HPO_4_; ≥99.0%), and trypsin (from the porcine pancreas) were purchased from Sigma-Aldrich (Saint Louis, MO, USA). Resazurin sodium salt (≥75%) was purchased from VWR International (Radnor, PA, USA). Dulbecco’s modified Eagle’s medium (DMEM, 11054) was purchased from Thermo Fisher Scientific (Waltham, MA, USA).

### 2.3. Syntheses of AuBSA and AuBSA-Fe—Their Purification, Concentrate Formation, and Storage

The synthetic procedure of AuBSA system follows the one used in our previous manuscript [22]. Briefly, DI water (0.2 mL) was added to an aqueous HAuCl_4_ solution (0.8 mL, 12.5 mM) and, subsequently, BSA solution (1 mL, 1 mM) was introduced under vigorous stirring (600 rpm). After 90 s, NaOH solution (0.2 mL, 1 M) was added to obtain a basic environment (pH ≈ 12). Ninety seconds later, the mixed solution was heated up in a microwave oven for 10 s (power was set to 150 W). The preparation of the AuBSA-Fe system differs only in the gradual addition of DI water (0.05 mL), FeCl_2_ (0.05 mL, 5 mM) and FeCl_3_ (0.1 mL, 5 mM) to an aqueous HAuCl_4_ solution instead of DI water volume (0.2 mL) alone.

After two hours of maturing at room temperature, the samples were dialyzed with a 14 kDa cut-off dialysis membrane (regenerated cellulose, Membra-Cel^TM^) against DI water. Dialysis was performed at room temperature for 24 h, with DI water being changed twice: once after the first hour and then again after the second hour. Concentrated forms of samples were prepared using a centrifugal concentrator (30 kDa). The rotational speed was set to 5000 rpm and the centrifugation lasted for 5 min. This process was performed repeatedly until the desired concentration was reached. Dialyzed and concentrated samples were stored in the dark at 4 °C.

### 2.4. Characterization Techniques

#### 2.4.1. Fluorescence Spectroscopy 

The fluorescence measurements of AuBSA and AuBSA-Fe systems were performed on a JASCO F8500 (Jasco, Tokyo, Japan) spectrofluorometer using a 1 cm quartz cuvette. Excitation–emission 3D maps were measured in the excitation range of 250–850 nm with a data interval of 5 nm and in an emission range of 250–850 nm with a data interval of 1 nm and a scan speed of 5000 nm·min^−1^. Emission spectra were measured in the range of 500–850 nm with a data interval of 1 nm and a scan speed of 100 nm·min^−1^. The excitation wavelength was set to 480 nm. All spectra were corrected to avoid any deviations induced by instrumental components.

The quantum yield of fluorescence (*Φ*) was then calculated by Equation (1):(1)$$\Phi =\Phi_{s}\cdot \frac{F\cdot \left(1-10^{-A_{s}}\right)\cdot n^{2}}{F_{s}\cdot \left(1-10^{-A}\right)\cdot n_{s}^{2}},$$
where *F* is the integrated fluorescence intensity, *A* is the absorbance, *n* is the index of refraction, and the subscript s indicates the standard. DCM, 4-(dicyanomethylene)-2-methyl-6-(4-dimethylaminostyryl)-4H-pyran, dissolved in ethanol (99.8%, Lach-Ner, Neratovice, Czech Republic) was used as a standard (*Φ*_s_ = 0.437 ± 0.024) [38]. 

Absorbance was measured on a Specord 250 Plus—223G1032 (Analytik Jena, Jena, Germany) with a double beam arrangement using a 1 cm quartz cuvette. As a reference, a 1 cm quartz cuvette filled with DI water was used. 

The hydrodynamic diameter of both systems was determined by dynamic light scattering using Zetasizer Nano ZS (Malvern Instruments Ltd., Malvern, UK) equipped with a He-Ne laser (λ = 633 nm) at 22 ± 1 °C. For fluorescence, absorbance, and hydrodynamic diameter measurements, the ratio of the sample dilution with DI water was the same.

#### 2.4.2. HR-TEM, STEM, and EDS

The AuBSA-Fe samples were measured by HR-TEM Titan G2 60–300 (FEI, Hillsboro, OR, USA) with an image corrector at an accelerating voltage of 300 kV. Images were taken with a BM UltraScan CCD camera (Gatan, Pleasanton, CA, USA). Energy Dispersive Spectrometry (EDS) was performed in STEM mode by a Super-X system with four silicon drift detectors (Bruker, Billerica, MA, USA). STEM images were taken with an HAADF detector 3000 (Fishione, Export, PA, USA).

#### 2.4.3. Mössbauer Spectroscopy

A home-made Mössbauer spectrometer was used to determine the oxidation and spin state of iron atoms within AuBSA-Fe samples. A representative as-prepared and centrifuged AuBSA-Fe sample was measured with an OLTWINS Mössbauer spectrometer in the transmission mode [39], using a constant acceleration rate and ^57^Co (Rh) source. The isomer shift values were related to the 28 μm α-Fe foil (Ritverc) measured at room temperature. By using measurements in magnetic field at low temperature, average sizes of SPIONs within AuBSA-Fe samples could be roughly estimated. The acquired Mössbauer spectra were processed using MossWinn 4.0 software [40].

#### 2.4.4. XPS

The X-ray photoelectron spectroscopy (XPS) measurements were carried out with the PHI 5000 VersaProbe II XPS system (Physical Electronics) with a monochromatic Al-Ka source (15 kV, 50 W) and a photon energy of 1486.7 eV. All the spectra were measured in a vacuum of 1.1 × 10^−7^ Pa and at a room temperature of 20 °C. Dual beam charge compensation was used for all measurements. The spectra were evaluated with MultiPak software, version 9 (Ulvac—PHI, Inc., Chanhassen, MN, USA).

#### 2.4.5. ICP-MS

To accurately determine the total Au and Fe concentrations, the validated ICP-MS method was employed. Prior to ICP-MS analysis, each sample was sonicated and consequently digested using MLS 1200 mega closed vessel microwave digestion unit (Milestone, Italy). The organic matrix was decomposed by a mixture of 4 mL of nitric acid (69%, Analpure) and hydrochloric acid (36%, Analpure) in 1:1 ratio. The digests were allowed to cool down to laboratory temperature, diluted with the ultrapure water to 25 mL in volumetric flasks, and stored at 4 °C until ICP-MS analysis. The detailed ICP-MS method description and the corresponding validation in terms of the limit of detection (LOD), the limit of quantification (LOQ), trueness, and precision are presented in the Appendix A. All ICP-MS measurements were performed in six replicates, and the results are expressed as an average ± standard deviation (SD).

#### 2.4.6. MR Relaxometry and Imaging

The MR relaxometry was used to determine the relaxivities r_1,2_ of AuBSA-Fe nanocomposites (M1–M4). The relaxation times T_1_ and T_2_ were measured on relaxometer Bruker Minispec mq 60 (Bruker Biospin, Ettlingen, Germany) at 1.5 T, at a stabilized temperature of 37 °C throughout the whole experiment. MR sequence for T_1_ measurement: Inversion recovery (IR), 20 points for fitting, 1 excitation, time of repetition (TR) = 0.01–10,000 ms, recycle delay 2 s. T_2_ relaxation times were measured with Carr-Purcell-Meiboom-Gill (CPMG), TR = 5000 ms, 20,000 echoes, 1 excitation, echo time (TE) = 0.05 ms, recycle delay 2 s. The relaxivities r_1,2_ were calculated via the least-squares curve fitting of R_1_ and R_2_ relaxation rates [s^−1^] versus iron concentration (mM). The experimentally determined solvent relaxation rate R was subtracted as a starting value from the nanoparticle relaxation rates prior to the linear regression analysis. 

The MR imaging experiments were performed on a Bruker Biospec 47/20 (Bruker, Ettlingen, Germany) at 4.7 T. T_1_- and T_2_-weighted MR images of M1–M4 and water (served as a control) samples in tubes were acquired. Rapid acquisition with relaxation enhancement (RARE) multi-spin echo MR sequence were used with the following parameters: T_1_-weighted sequence: effective echo time (TE) = 11.6 ms, time of repetition (TR) = 587.0 ms, turbo factor (TF) = 1, scan time = 10.5 min, plane resolution (PR) = 234 × 195 µm^2^, slice thickness = 0.6 mm. T_2_-weighted sequence: RARE, TE = 36 ms, TR = 3300 ms, TF = 8, scan time = 11.0 min, PR = 234 × 195 µm^2^, slice thickness = 0.6 mm. MR image processing and quantification were performed using ImageJ software. The signal-to-noise ratio (SNR) was calculated from images as 0.655 × S_sample_/σ_noise_ and contrast-to-noise ratio (CNR) was calculated from images as 0.655 × |S_sample_ − S_water_|/σ_noise_, where S is signal intensity in the region of interest, σ is the standard deviation of background noise, and the constant 0.655 reflects the Rician distribution of background noise in a magnitude MR image.

#### 2.4.7. Alamar Blue Assay (Resazurin Assay)

In a typical experiment, 80 μL of cultivation medium (second column) or cell (RPE-1) suspension was added to a 96-well plate, which was afterward plaved inside the incubator (37 °C, 5% CO_2_). After 24 h, 20 μL of DI water (second and third column), two different concentrations of gold and iron precursors, or AuBSA-Fe nanocomposites were added in the form of tri/hexaplicates. Another 24 h later, 20 μL of resazurin was introduced to each well. After 3 h of incubation, fluorescence intensity was measured on a microplate reader Synergy Mx (BioTek^TM^, Winooski, VT, USA). The excitation and emission wavelengths were set to 540 nm and 590 nm, respectively. Cell viability (*CV*) was calculated according to Equation (2):(2)$$CV=100\times \frac{F_{\mathrm{sample}}-F_{\mathrm{medium}}}{F_{\mathrm{cells}}-F_{\mathrm{medium}}},$$
where *F* is the averaged fluorescence intensity and the subscripts sample, cells, and medium indicate the measurement of fluorescence in the suspensions of sample-treated cells, non-treated cells, and the solution of cultivation medium alone, respectively.

## 3. Results and Discussion

The samples of AuBSA-Fe were prepared by an easy one-pot synthetic procedure performed in a sequential manner, which was newly developed by us, as described in detail in the Materials and Methods section. Essentially, ferrous and ferric ions were mixed together in the ratio of 1:2, added to Au (III) aqueous solution and then allowed to interact with BSA for a certain period. The reaction mixture was alkalized in the next step to set up conditions for simultaneous and spontaneous Au (III) reduction and SPIONs formation (i.e., precipitation of Fe ions under alkaline medium in the presence of BSA); the subsequent heating accelerated both bio-mineralization reactions. As a reference, the AuBSA sample was prepared by using the same amount of Au (III) and BSA as in AuBSA-Fe system. Thus, AuBSA and AuBSA-Fe systems differ only in the absence/presence of iron ions in their synthetic procedures, respectively. The procedures of both nanocomposite syntheses are schematically depicted in Figure 1.

### 3.1. Luminescent Properties of AuBSA-Fe in Comparison to AuBSA

There might be concerns about luminescence quenching induced by iron cations, since luminescent AuNCs have been used as sensors of Fe (III) in solution [41,42,43]. However, in the cited studies, BSA is not used as the template for luminescent AuNCs formation. Moreover, there is a big difference between (i) Fe cations being present in the course of luminescent AuNCs formation within BSA (this study) and (ii) Fe cations being added to well-formed luminescent AuNCs [41,42,43].

Prompted by this issue, we first focused our attention on the validation of luminescent properties of AuNCs in the AuBSA-Fe system inherited from AuBSA—see Figure 2 for emission spectra in the region of 500–850 nm and Appendix A for the whole-range 3D excitation–emission maps. Obviously, the average position of the emission maximum of AuNCs remained almost unchanged when iron ions were present: 657 ± 2 nm for AuBSA and 656 ± 1 nm for AuBSA-Fe (Appendix A, respectively). The intensity of luminescence decreased slightly in AuBSA-Fe in comparison to AuBSA (Figure 2). The fluorescent quantum yield reflects this fact and is of virtually the same average value for AuBSA-Fe, 6.2 ± 0.2 (Appendix A), as for AuBSA, 6.4 ± 0.1 (Appendix A). This is a good sign that qualitative and quantitative luminescent features of AuNCs are not affected by the presence of iron atoms in AuBSA-Fe samples. Furthermore, one can assume that sizes and numbers of AuNCs within AuBSA-Fe and AuBSA nanocomposites are approximately the same.

### 3.2. Investigation of Morphology and Particle Size Distribution in Luminescent AuBSA-Fe

According to STEM image in Figure 3A, one can see relatively large aggregates exceeding several hundreds of nanometres in size; however, they consist of individual particles with sizes in units of nanometres and are frequently encountered in AuBSA-Fe systems. EDS data shown in Figure 3B,C further demonstrate that oxygen dominates in the close vicinity of iron in nanoparticulate form (e.g., Fe_x_O_y_), while sulphur can be co-located together with gold atoms, respectively. This supports previous results of many researchers (including us, [22]) concerning Au–S interactions within AuBSA. It also correlates well with the observation that the luminescent features of AuNCs are not severely hampered by the presence of Fe_x_O_y_ in AuBSA-Fe. Thus, we anticipate that the same type of amino-acid residues creates the closest nano-environment of luminescent AuNCs in AuBSA-Fe as that in AuBSA systems. Since the samples for STEM/EDS are prepared by drying on a support (lacey carbon-coated Cu grid), the real particle size distribution (PSD) in the solution may differ from that observed by STEM. Therefore, it is reasonable to determine PSD directly by measuring the aqueous solutions of the samples by DLS. The average values of the hydrodynamic diameters of particles in AuBSA and AuBSA-Fe nanocomposite solutions along with polydispersity values (PDI) determined by DLS are compared in Table 1. Both samples (AuBSA as well as AuBSA-Fe) represent proper solutions without any obvious aggregate formation visible by the naked eye.

Obviously, both the hydrodynamic diameter and PDI increased in AuBSA-Fe in comparison to AuBSA (Table 1). These increases in the average values (from approx. 24 nm in diameter and 0.4 polydispersity in AuBSA to 71 nm and 1.0 in AuBSA-Fe) can be ascribed to the presence of iron oxide particles and their aggregates in AuBSA-Fe because these are the only differences between the two compared systems. Further details of DLS data are shown and discussed in the Appendix A; whereas appropriate values for AuBSA and AuBSA-Fe are listed in Appendix A, respectively. Although influenced by sample drying to some extent, the STEM images of AuBSA-Fe in dried state (Figure 3A) correlate with the PSD determined for the same system by DLS measured directly in aqueous solution (liquid state).

### 3.3. Evidence of SPIONs in Luminescent AuBSA-Fe via Mössbauer Spectroscopy

Mössbauer spectroscopy as an iron-sensitive method has been selected to give direct evidence regarding the type of iron oxide present in AuBSA-Fe. Since relatively high concentrations of iron are required in this spectroscopy and, simultaneously, by knowing (from STEM-EDS) that iron is most dominantly distributed in nanoparticulate form at the surface of BSA, we centrifuged the AuBSA-Fe samples, a rusty pellet was carefully dried under nitrogen atmosphere and then measured. The Mössbauer spectrum of AuBSA-Fe recorded at room temperature, shown in Figure 4A, manifested itself as a doublet with an isomer shift value of 0.33 ± 0.01 mm·s^−1^ and the quadrupole splitting of 0.68 ± 0.01 mm·s^−1^. By measuring the Mössbauer spectrum at 5 K and 5 T, as seen Figure 4B, a sextet with an isomer shift value of 0.43 ± 0.01 mm·s^−1^, a quadrupole splitting of −0.08 ± 0.01 mm·s^−1^, and an effective hyperfine magnetic field of 46.4 ± 0.3 T was revealed. Based on these parameters and our previous knowledge [44], the nanoparticulate form of iron in AuBSA-Fe samples can be assigned to superparamagnetic Fe (III) oxide. Furthermore, the measurements at low temperatures and under external magnetic fields showed a symmetrical environment with no preferential orientation; therefore, very small superparamagnetic iron oxide particles (SPIONs) are present in AuBSA-Fe, generally in units of nanometres. This coincides well with STEM imaging and DLS analysis.

### 3.4. Application of Luminescent AuBSA-Fe as MRI Contrast Agents

SPIONs are well-known as negative or T_2_-weighted MRI contrast agents [23]. Therefore, we assessed MRI performance of our AuBSA-Fe samples. In Figure 5, we show the T_2_-weighted MR images of four independently prepared AuBSA-Fe samples (denoted as M1–M4), containing different (increasing) concentrations of gold and iron, as determined by ICP-MS (Appendix A), but keeping the same molar ratio of these metals (10:0.75). Intentionally, four independently prepared samples were concentrated to verify the reproducibility and to increase the T_2_-weighted signal. Obviously, the T_2_-weighted MR images of water phantoms were affected by the presence of AuBSA-Fe samples and the clear decrease in the MR signal was observed; on the other hand, and as expected, the only negligible effect was observed in T_1_-weighted MR images, where the MR signal increase was low (Figure 5). The values of signal-to-noise ratio (SNR) as well as contrast-to-noise ratio (CNR) for the quantitative comparison of M1–M4 samples are listed in Figure 5. Both SNR and CNR reached values above 37 in the T_2_-weighted MR images of all four variants of AuBSA-Fe samples; simultaneously, low SNR and CNR values in T_1_-weighted MR images were achieved. This means that AuBSA-Fe samples represent “negative” contrast agents due to the presence of SPIONs. This is in full accordance with the literature [29,30].

Aside from the MRI imaging of water phantoms containing AuBSA-Fe samples (M1–M4), MR relaxometry was performed. The relaxation rates R_1_ and R_2_ were calculated as 1/T_1_ and 1/T_2_, respectively, for concentrated and diluted M1–M4 samples. Note that the real concentrations of Fe in concentrated and diluted M1–M4 samples together with the corresponding values of R_1_, R_2_ relaxation rates are listed in Appendix A. Plotting the relaxation rates as a function of real iron concentration in AuBSA-Fe samples (determined by ICP-MS) resulted in the determination of relaxivities r_1_ and r_2_ from graphs shown in Figure 6. Evidently, the experimental R_1_ values could be best fitted with a linear function (although it can be separated in two parts, according to R_2_ dependence). On the other hand, two linear functions with two different slopes are best able to fit the experimental R_2_ values: 3.44 ± 0.36 L∙mmol^−1^∙s^−1^ for iron concentrations equal and above 0.52 mM; 2.68 ± 0.11 L∙mmol^−1^∙s^−1^ for iron concentrations below this value (Table 2). These slopes represent the characteristic r_2_ relaxivity of AuBSA-Fe samples and, as such, can be compared with the relaxivity values of the commercial MRI contrast agents (e.g., in [45]). From this direct comparison, it is obvious that the r_2_ relaxivity values of AuBSA-Fe samples closely approach those of several commercially available contrast agents. Importantly, the commercial MRI contrast agents do not possess luminescent properties, while AuBSA-Fe samples do. Therefore, AuBSA-Fe samples could serve as bimodal (dual) probes for MRI and fluorescence measurements.

Interestingly, R_2_ values may be even fitted with a quadratic function as shown for concentrated samples in Appendix A. The quadratic dependence of relaxation rates on concentration of contrast agents was observed in previous studies by different authors [46,47,48,49,50,51]. In our opinion, two plausible explanations may be adopted in the case of AuBSA-Fe samples: either the aggregation of SPIONs and the consequent inhomogeneity of magnetic fields as in [52] or the small sizes of SPIONs (evidenced for our AuBSA-Fe samples through direct visualization using STEM and/or spectroscopically through the Mössbauer effect), thus falling in a range of quadratic relaxation [53].

### 3.5. Stability and Biocompatibility of AuBSA-Fe Nanocomposites

An important issue in any sample applicability is their stability in time if stored under relevant conditions. Since AuBSA-Fe samples contain inorganic parts, being responsible for luminescent and MR features as well as protein (although denatured during the synthesis), generally, we stored our samples in a fridge. However, for the sake of curiosity, a sample of AuBSA-Fe was stored at room temperature over 1 year, and its X-ray photoelectron spectroscopic (XPS) spectrum measured and directly compared with that of freshly prepared AuBSA-Fe. The XPS results, shown in Appendix A and discussed in the Appendix A, confirmed the degradation of organic part, while preserving Au (0) content even in the AuBSA-Fe sample stored at room temperature. Thus, the stability of the newly developed AuBSA-Fe dual probes was verified. It can be summed up that AuBSA-Fe, representing a stable system when stored in a fridge, could potentially be applied as fluorescent and MRI contrast agents. 

Another very important issue of AuBSA-Fe nanocomposite application as a potential contrast agent is its biocompatibility. Since AuBSA-Fe nanocomposites are prepared by a synthetic approach obeying the principles of green chemistry (i.e., non-toxic reactants and aqueous environments, no abundant chemicals used), their biocompatibility can be presumed. Moreover, AuBSA nanocomposites have been tested by many authors, including us [19], for potential cytotoxicity, which was revealed to be negligible. Similarly, SPIONs were tested by several authors and manifested almost zero cytotoxicity (e.g., [25,26,27,28]). It would be thus very unusual if AuBSA-Fe nanocomposites were cytotoxic. However, the assumption of the low cytotoxicity of AuBSA-Fe was validated by using Alamar blue assay (exploiting resazurin and fluorescence measurements) in the present study. The average cell viabilities for AuBSA-Fe nanocomposites with different iron concentrations (below and/or above 0.52 mM Fe content, in correlation with MRI data) are shown in Table 3, and an example of the resazurin assay is given in Appendix A. 

Surprisingly, the average cell viability was determined to be around 80% (only) in all AuBSA-Fe nanocomposites. This value still falls in the range of non-toxic species according to ISO 10993. However, it should be pointed out that the cytotoxicity results may be false negatives because resazurin is able to interact with serum albumin, especially at elevated protein concentrations, as revealed in [54]. In which case, the final values of cell viability (here evaluated around 80%) could be underestimated with respect to reality, i.e., the biocompatibility of AuBSA-Fe nanocomposites could be much better than determined by the Alamar blue assay. It should be also noted that the MTT assay and CCK-8 kit were not employed because both are able to provide false-positive results, as discussed in [55,56]. Further experiments assessing the real cytotoxicity of AuBSA-Fe nanocomposites are in progress.

## 4. Conclusions

We developed an easy, reproducible, one-pot, green synthesis of a new type of potential bimodal probe, labelled as AuBSA-Fe. These AuBSA-Fe probes are based on non-toxic luminescent AuNCs (embedded in BSA), which are generated together with SPIONs simply through the alkalization of the reaction mixture. Luminescent features of AuNCs are preserved in AuBSA-Fe samples, i.e., emission maxima and quantum yields are comparable within experimental errors with those of AuBSA (serving here as a reference). Furthermore, MRI experiments confirmed the effect of AuBSA-Fe on T_2_ contrast in MR images. The relaxivity values of AuBSA-Fe approach those of commercial contrast agents. The great benefit of AuBSA-Fe probes, serving as MR alternatives, lies in their simultaneous luminescent feature. Therefore, AuBSA-Fe nanocomposites (stable when stored in a fridge) represent promising bimodal probes and could be potentially applied as fluorescent and MRI contrast agents. Further experiments with AuBSA-Fe nanocomposites are envisaged, leading to the increased possibility of their use as MRI alternatives and testing their biocompatibility and stability, performed not only *in vitro* but also *in vivo*.

## Figures and Tables

**Figure 1 nanomaterials-13-01027-f001:**
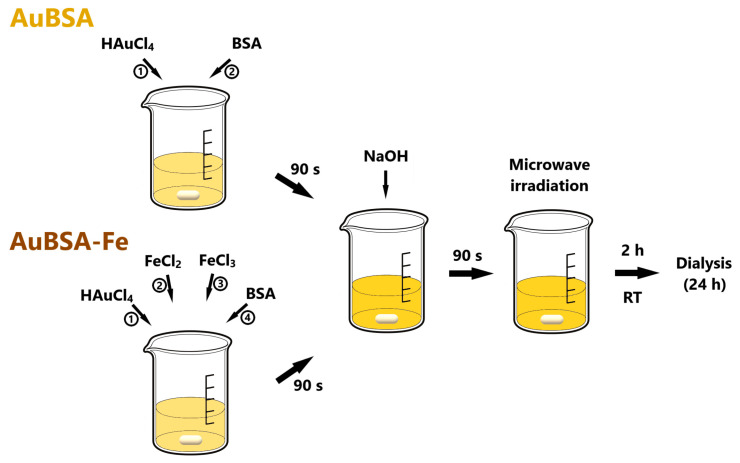
Schematic depiction of AuBSA and AuBSA-Fe nanocomposites.

**Figure 2 nanomaterials-13-01027-f002:**
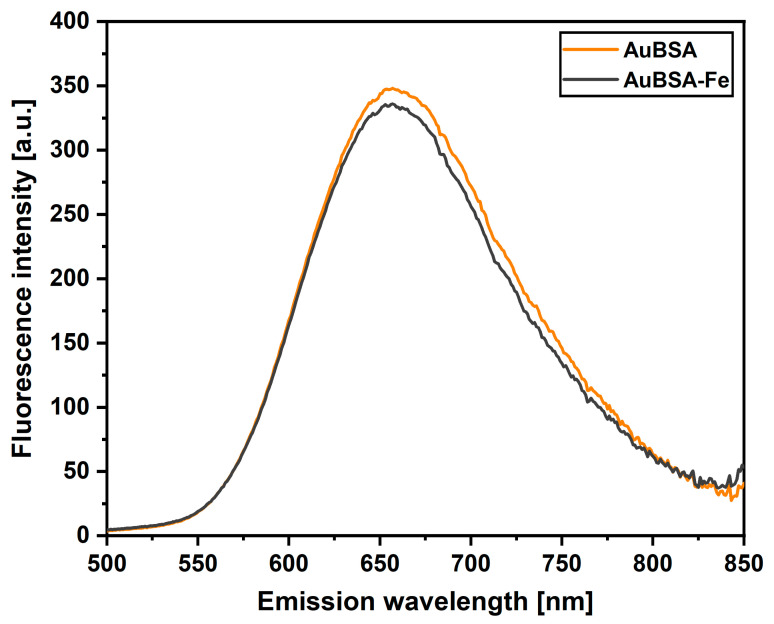
Comparison of fluorescence emission spectra of AuBSA (orange curve) and AuBSA-Fe (black curve) samples when excited at 480 nm. Average fluorescence spectra are shown as a result of seven independent sample preparations and their measurements.

**Figure 3 nanomaterials-13-01027-f003:**
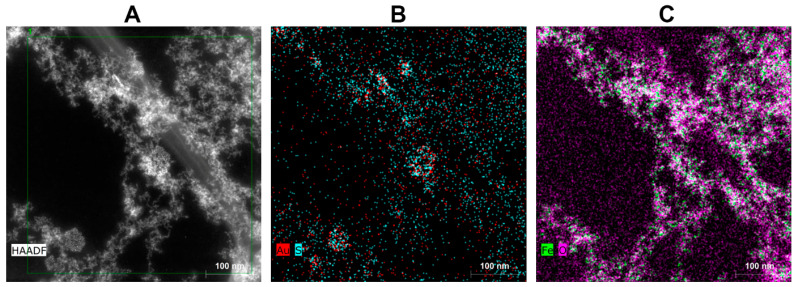
Scanning transmission electron microscopy (STEM) and energy dispersive spectroscopic (EDS) analysis of a representative AuBSA-Fe sample: (**A**) STEM image, scale bar of 100 nm; (**B**) Au and S spatial distribution map; and (**C**) Fe and O spatial distribution map.

**Figure 4 nanomaterials-13-01027-f004:**
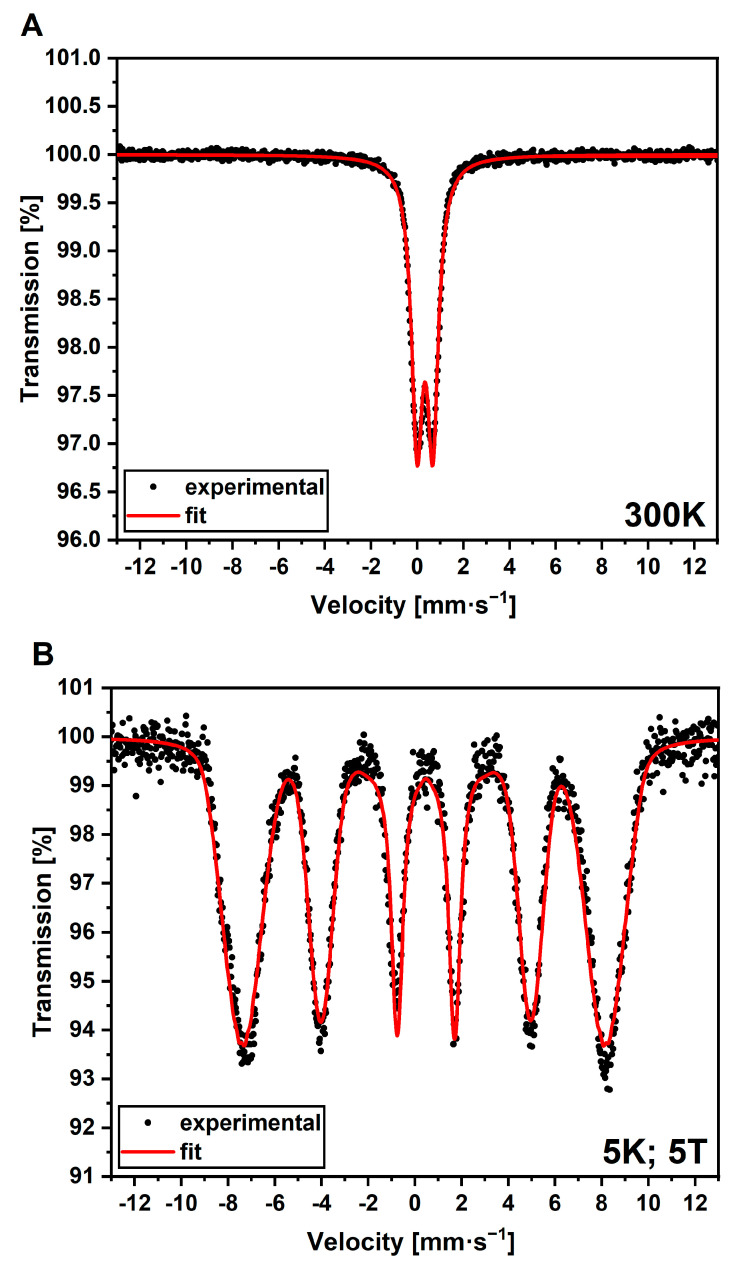
Mössbauer spectra of AuBSA-Fe recorded (**A**) at room temperature and (**B**) at the 5 K and 5 T external magnetic field.

**Figure 5 nanomaterials-13-01027-f005:**
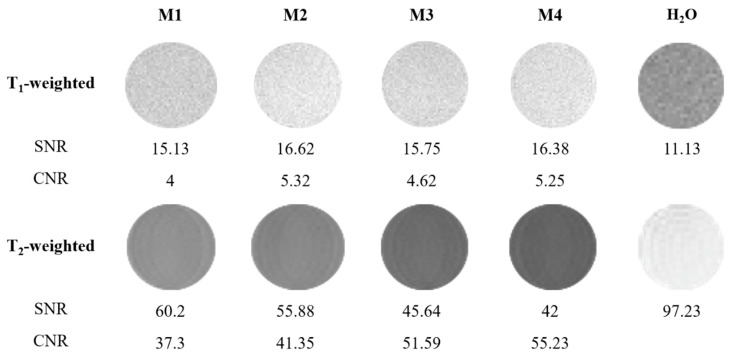
Magnetic resonance (MR) images of AuBSA-Fe phantoms (denoted as M1–M4) containing different Fe concentrations (807 µM Fe in M1, 1020 µM Fe in M2, 1193 µM Fe in M3, and 1249 µM Fe in M4) and water phantom (H_2_O), measured at 4.7 T external magnetic field. T_1_- and T_2_-weighted MR images are shown. Note: the signal-to-noise ratio (SNR) was calculated using SNR = 0.655 × S/σ, where S is signal intensity in the region of interest (ROI), σ is the standard deviation of background noise, and the constant 0.655 reflects the Rician distribution of background noise in a magnitude MR image. Eight averages were used.

**Figure 6 nanomaterials-13-01027-f006:**
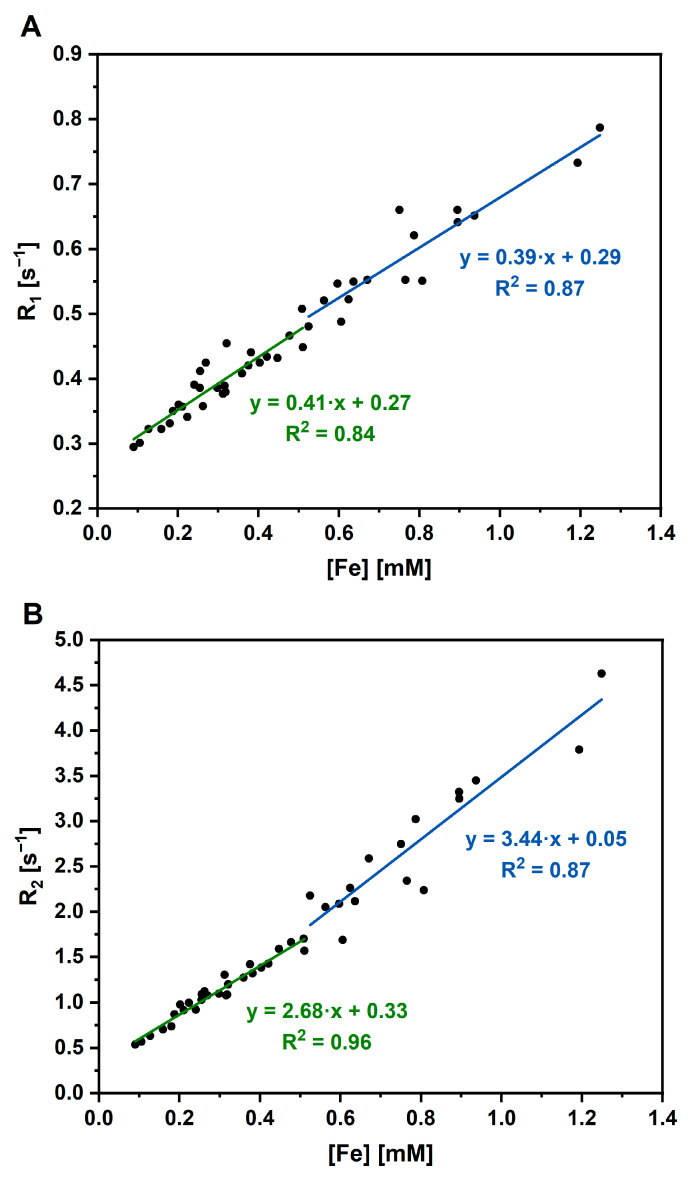
(**A**) Relaxation rate R_1_ and (**B**) relaxation rate R_2_ as functions of real iron concentrations in AuBSA-Fe samples, as determined by ICP-MS (values are listed in Appendix A).

**Table 1 nanomaterials-13-01027-t001:** Hydrodynamic diameter (represented by Z-average) and polydispersity (PDI) of nanocomposites determined by DLS measurements.

Sample	Z-Average [nm]	PDI
AuBSA	23.9 ± 10.8	0.4 ± 0.1
AuBSA-Fe	71.2 ± 8.0	1.0 ± 0.0

**Table 2 nanomaterials-13-01027-t002:** The values of r_1_ and r_2_ relaxivities assessed for AuBSA-Fe samples, depending on real iron concentrations, as determined by ICP-MS. Two linear regimes are recognized by a jump around the value of 0.52 mM in Fe concentrations.

Fe Concentration [mM]	Relaxivity r_1_ [L·mmol^−1^·s^−1^]	Relaxivity r_2_ [L·mmol^−1^·s^−1^]
<0.52	0.41 ± 0.04	2.68 ± 0.11
≥0.52	0.39 ± 0.04	3.44 ± 0.36

**Table 3 nanomaterials-13-01027-t003:** Results of cell viability tests.

AuBSA-Fe	Average Viability [%]
Iron concentration < 0.52 mM	78 ± 3
Iron concentration ≥ 0.52 mM	80 ± 2

## Data Availability

The raw/processed data required to reproduce these findings can be requested from the authors directly.

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
