# Peer review of "Facile One-Pot Green Synthesis of Magneto-Luminescent Bimetallic Nanocomposites with Potential as Dual Imaging Agent"

_nanomaterials, 2023, doi:10.3390/nano13061027_

Round 1
Reviewer 1 Report
Ostruszka and co-workers reported the title of “Magneto-luminescent bimetallic nanocomposites applicable as dual imaging probes”, which the experimental is carefully conducted, and the results have been presented correctly, and the contents fall well into the scope of the journal. I recommend the publication of this paper after minor revision :
1. The unit “ml” should be changed to “mL”. Please check the whole paper and correct them.
2. What does the mean of ‘Mössbauer’ in 2.3.3 Mössbauer spectroscopy?
3. What is the biocompatibility of AuNCs-BSA-SPIONs? As the compound is used for MRI, which the biocompatibility is an unavoidable problem.
4. I strongly suggest the authors read through the paper and correct such typographical errors.
Author Response
We thank to all five reviewers for their valuable comments, suggestions, and questions. It greatly improved our manuscript that is now submitted in its revised version. Please, find below our responses made step-by-step to each reviewer comment.
All changes are also marked in the revised version of our manuscript by using track & trace mode and all revisions shown. The numbering of pages and lines in our responses to reviewers is related to the revised version of our manuscript with all revisions shown.
Reviewer 1
Open Review
( ) I would not like to sign my review report
(x) I would like to sign my review report
English language and style
( ) English very difficult to understand/incomprehensible
( ) Extensive editing of English language and style required
( ) Moderate English changes required
( ) English language and style are fine/minor spell check required
(x) I don't feel qualified to judge about the English language and style
|
Yes |
Can be improved |
Must be improved |
Not applicable |
|
|
Does the introduction provide sufficient background and include all relevant references? |
(x) |
( ) |
( ) |
( ) |
|
Are all the cited references relevant to the research? |
(x) |
( ) |
( ) |
( ) |
|
Is the research design appropriate? |
(x) |
( ) |
( ) |
( ) |
|
Are the methods adequately described? |
(x) |
( ) |
( ) |
( ) |
|
Are the results clearly presented? |
(x) |
( ) |
( ) |
( ) |
|
Are the conclusions supported by the results? |
(x) |
( ) |
( ) |
( ) |
Comments and Suggestions for Authors
Ostruszka and co-workers reported the title of “Magneto-luminescent bimetallic nanocomposites applicable as dual imaging probes”, which the experimental is carefully conducted, and the results have been presented correctly, and the contents fall well into the scope of the journal. I recommend the publication of this paper after minor revision :
- The unit “ml” should be changed to “mL”. Please check the whole paper and correct them.
Our response: We apologize for such a scholar mistake on several places in Experimental section. It was checked and corrected in the whole manuscript.
- What does the mean of ‘Mössbauer’ in 2.3.3 Mössbauer spectroscopy?
Our response: We included several sentences into this section to clarify the usage of a home-made Mossbauer spectrometer. The text of section 2.4.3. (previously 2.3.3.) now reads: „ Home-made Mössbauer spectrometer was used to determine the oxidation and spin state of iron atoms within AuBSA-Fe samples. A representative as-prepared and centrifuged AuBSA-Fe sample was measured with OLTWINS Mössbauer spectrometer in transmission mode [39] using a constant acceleration rate and 57Co (Rh) source. The isomer shift values were related to the 28 μm α-Fe foil (Ritverc) measured at room temperature. By using measurements in magnetic field at low temperature, average sizes of SPIONs within AuBSA-Fe samples could be estimated roughly. The acquired Mössbauer spectra were processed using MossWinn 4.0 software [40].“
- What is the biocompatibility of AuNCs-BSA-SPIONs? As the compound is used for MRI, which the biocompatibility is an unavoidable problem.
Our response: We intended to perform MTT assay, which is the most frequently used assay in cytotoxicity evaluation. However, MTT assay can overestimate the viability of cells when serum albumin present, leading thus to false-positive results as warned in (Funk D. et al., Serum albumin leads to false-positive results in the XTT and the MTT assay, Biotechiques 2007, 43, 178-182). Furthermore, in (Neufeld B. H. et al., Small molecule interferences in resazurin and mtt-based metabolic assays in the absence of cells, Analytical Chemistry 2018, doi: 0.1021/acs.analchem.8b01043), it is shown that significant conversion of the assay reagents (dyes) occurs in the presence of many small molecules mimicking thus cellular activity in systems with no cells present at all. Thus, great care must be taken in viability tests evaluation.
Generally, it is better to use fluorescence, rather than absorbance as for spectroscopic technique, due to the higher sensitivity of the former. Therefore, we rather choose Alamar blue assay (exploitation of resazurin and fluorescence measurements) for our cytotoxicity determination in the present study, similarly as we did in our previous paper concerning AuNCs-BSA biocompatibility assessment (Andrýsková et al., Nanomaterials 2020). The results of cell viability for AuNCs-BSA-SPIONs are now included in our manuscript on pages 13-14, Table 3. It should be, however, pointed out that the results may be still false-negative because resazurin is able to interact with serum albumin, especially at elevated protein concentrations as discussed in (Goegan P. et al., Effects of serum protein and colloid on alamar blue assay in cell cultures,Toxic. In Vitro 1995, 9, 257-266). The results of cell viability can be thus underestimated and need to be verified further by another assay. Indeed, it is the reason why we did not include it in our submitted version of the manuscript. However, prompted by several reviewers, we decided to show and comment it in our revised manuscript. Therefore, it can be found out in the revised version on pages 13-14, lines 454-484.
- I strongly suggest the authors read through the paper and correct such typographical errors.
Our response: We went through the whole manuscript and tried to correct all visible typographical errors.
Submission Date
17 February 2023
Date of this review
21 Feb 2023 15:46:53
Faithfully yours,
Assoc. Prof. Karolína Šišková
On behalf of the Authors
Dept. of Experimental Physics, Faculty of Science, Palacký University Olomouc, CZ karolina.siskova@upol.cz
Reviewer 2 Report
Radek and co-author synthesized Magneto-luminecent bimetallic nano composites applicable as dual imaging probes. The propose is interesting. However, it need more experiments demonstration this probes are suitable to use as bimodal probes. Here are my comments:
1, In the Figure 3. The size distribution is large. The author should measure the PDI. to check whether the sample is table to use as bio-probes.
2, In the Figure 5. The author should clearly show the different content of the Fe in the AuBSA-Fa of each independently experiments before taking the T2-weighted imaging. The signal is different on these four samples, the author should explain the reason and reproducible of the samples.
3, For further demonstrate the probes. The author should measure the properties appellation these probes in cell levels and in vivi and test whether it suitable to use as bioprocess. Without further data, it's hard to evaluate the probe properties for the application.
Author Response
We thank to all five reviewers for their valuable comments, suggestions, and questions. It greatly improved our manuscript that is now submitted in its revised version. Please, find below our responses made step-by-step to each reviewer comment.
All changes are also marked in the revised version of our manuscript by using track & trace mode and all revisions shown. The numbering of pages and lines in our responses to reviewers is related to the revised version of our manuscript with all revisions shown.
Reviewer2
Open Review
(x) I would not like to sign my review report
( ) I would like to sign my review report
English language and style
( ) English very difficult to understand/incomprehensible
( ) Extensive editing of English language and style required
( ) Moderate English changes required
( ) English language and style are fine/minor spell check required
(x) I don't feel qualified to judge about the English language and style
|
Yes |
Can be improved |
Must be improved |
Not applicable |
|
|
Does the introduction provide sufficient background and include all relevant references? |
(x) |
( ) |
( ) |
( ) |
|
Are all the cited references relevant to the research? |
( ) |
(x) |
( ) |
( ) |
|
Is the research design appropriate? |
( ) |
(x) |
( ) |
( ) |
|
Are the methods adequately described? |
( ) |
(x) |
( ) |
( ) |
|
Are the results clearly presented? |
( ) |
(x) |
( ) |
( ) |
|
Are the conclusions supported by the results? |
( ) |
(x) |
( ) |
( ) |
Comments and Suggestions for Authors
Radek and co-author synthesized Magneto-luminecent bimetallic nano composites applicable as dual imaging probes. The propose is interesting. However, it need more experiments demonstration this probes are suitable to use as bimodal probes. Here are my comments:
1, In the Figure 3. The size distribution is large. The author should measure the PDI. to check whether the sample is table to use as bio-probes.
Our response: We reconsidered the presentation of our DLS data based on this comment. As recommended by reviewer, we rather present Z-average values (meaning average hydrodynamic diameter of particles) and polydispersity (PDI) of AuBSA and AuBSA-Fe nanocomposite solutions. Therefore, new Table 1 is included and discussed within the main text (page 8, lines 319-321).Since our previous Figure 3 (showing average PSD based on intensity) would be possibly misleading for readers, we moved it into Supporting Information (currently presented as Figure SI-2) and added detailed discussion and explanation of its presentation.
2, In the Figure 5. The author should clearly show the different content of the Fe in the AuBSA-Fa of each independently experiments before taking the T2-weighted imaging. The signal is different on these four samples, the author should explain the reason and reproducible of the samples.
Our response: We included iron concentrations (determined by ICP-MS and listed in Supporting Information in Table SI-6) in Figure caption of Figure 5. It now reads (page 11): „Figure 5. Magnetic resonance (MR) images of AuBSA-Fe phantoms (denoted as M1-M4) containing different Fe concentrations (807 µM Fe in M1, 1020 µM Fe in M2, 1193 µM Fe in M3, 1249 µM Fe in M4) and water phantom (H2O), measured at 4.7 T external magnetic field. T1- and T2-weighted MR images are shown. Note: The signal-to-noise ratio (SNR) was calculated using SNR= 0.655*S/σ, where S is signal intensity in the region of interest (ROI), σ is the standard deviation of background noise and constant 0.655 reflects the Rician distribution of background noise in a magnitude MR image. 8 averages were used.“
Simultaneously, we add the following sentence explaining the concentrate formation (M1-M4) in section 3.4 – lines 383-384: “Intentionally, four independently prepared samples were concentrated to verify the reproducibility and to increase the T2-weighted signal.”
3, For further demonstrate the probes. The author should measure the properties appellation these probes in cell levels and in vivi and test whether it suitable to use as bioprocess. Without further data, it's hard to evaluate the probe properties for the application.
Our response: Biocompatibility tests were included in our revised version of the manuscript (pages 13-14, section 3.5). In vivo tests are beyond the scope of the current study since our work focuses on development of synthesis of new nanocomposites and their thorough characterization with the aim to demonstrate their potential applicability as dual probes. Therefore, we softened the title (line 2, page 1) as well as the text in line 453 (page 13) by the word “potentially”, we also stressed the green synthesis within the title (line 3, page 1), changed “can” to “could” on page 12 (line 421), and partially reformulated the first sentence in Conclusions (page 14).
Submission Date
17 February 2023
Date of this review
22 Feb 2023 22:01:12
Faithfully yours,
Assoc. Prof. Karolína Šišková
On behalf of the Authors
Dept. of Experimental Physics, Faculty of Science, Palacký University Olomouc, CZ karolina.siskova@upol.cz

Reviewer 3 Report
The manuscript by Šišková et al. describes facile one-pot green synthesis of a new type of bimetallic nanocomposite (AuBSA-Fe) and claimed it as a bimodal probe for magnetic resonance as well as luminescent imaging application. Bimetallic nanoprobes has also been reported in past for similar dual imaging applications (Nanomaterials 2016, 6(4), 65, Biomater. Sci., 2017, 5, 2122). The main innovation of the current work is the use of Fe metal in nanocomposite instead of Gd. Although, the topic of research is appealing and interesting, authors need to clarify the following points before the publication of this manuscript-
Authors need to justify the use of Fe over other metals like Gd in the nanocomposite which is exhaustively reported in literature. It is worthy to mention that Gd/AuNC can also perform magnetic resonance and fluorescence imaging. Adding this information will strengthen the results and justify the proposed strategy.
It is claimed that the designed nanocomposite system AuBSA-Fe is applicable for magnetic resonance and luminescent imaging. However, no fluorescence imaging data is provided.
Authors need to perform MTT assay before coming to the conclusion that AuBSA-Fe nanocomposite is non-toxic.
Significant increment in the particle size of AuBSA from 266 to 351 nm is observed when AuBSA-Fe is formed. This strongly indicates the occurrence of aggregation. It is surprising why the fluorescence quantum yield is reduced only marginally from 6.4 to 6.2%, since Fe is a good quencher?
The manuscript need to be thoroughly checked by English native speaker as many sentences are grammatically incorrect and require appropriate modification (some are highlighted below).
In Abstract: “one-pot sequential green synthesis is developed”……. “By alkalization in the right moment”
On Page 1, Line 41: “with the first type of the approaches to”
On Page 5, Line 219-220: “The intensity of luminescence only very slightly decreased”
Author Response
We thank to all five reviewers for their valuable comments, suggestions, and questions. It greatly improved our manuscript that is now submitted in its revised version. Please, find below our responses made step-by-step to each reviewer comment.
All changes are also marked in the revised version of our manuscript by using track & trace mode and all revisions shown. The numbering of pages and lines in our responses to reviewers is related to the revised version of our manuscript with all revisions shown.
Reviewer 3
Open Review
(x) I would not like to sign my review report
( ) I would like to sign my review report
Quality of English Language
( ) English very difficult to understand/incomprehensible
(x) Extensive editing of English language and style required
( ) Moderate English changes required
( ) English language and style are fine/minor spell check required
( ) I am not qualified to assess the quality of English in this paper
|
Yes |
Can be improved |
Must be improved |
Not applicable |
|
|
Does the introduction provide sufficient background and include all relevant references? |
( ) |
(x) |
( ) |
( ) |
|
Are all the cited references relevant to the research? |
(x) |
( ) |
( ) |
( ) |
|
Is the research design appropriate? |
( ) |
( ) |
(x) |
( ) |
|
Are the methods adequately described? |
( ) |
(x) |
( ) |
( ) |
|
Are the results clearly presented? |
( ) |
(x) |
( ) |
( ) |
|
Are the conclusions supported by the results? |
( ) |
( ) |
(x) |
( ) |
Comments and Suggestions for Authors
The manuscript by Šišková et al. describes facile one-pot green synthesis of a new type of bimetallic nanocomposite (AuBSA-Fe) and claimed it as a bimodal probe for magnetic resonance as well as luminescent imaging application. Bimetallic nanoprobes has also been reported in past for similar dual imaging applications (Nanomaterials 2016, 6(4), 65, Biomater. Sci., 2017, 5, 2122). The main innovation of the current work is the use of Fe metal in nanocomposite instead of Gd. Although, the topic of research is appealing and interesting, authors need to clarify the following points before the publication of this manuscript-
Authors need to justify the use of Fe over other metals like Gd in the nanocomposite which is exhaustively reported in literature. It is worthy to mention that Gd/AuNC can also perform magnetic resonance and fluorescence imaging. Adding this information will strengthen the results and justify the proposed strategy.
Our response: We are aware of the fact that there are many studies dealing with nanocomposites consisting of Gd (III) complexes and luminescent part (either AuNCs, or fluorescent dyes) as we mention in Introduction. However, Gd (III) species are toxic and represent potential risk to environment and human health (Harini G. et al, Journal of Cleaner Production 2022, 345, 131139). Hence, instead of Gd (III) species, it is meaningful to exploit superparamagnetic iron oxide nanoparticles (SPIONs) which are non-toxic as proved by several authors (Arias L. S. et al, Antibiotics 2018, 7, 46; Bajaj A. et al., J. Mater. Chem. 2009, 19, 6328-6331; Li D. et al, Anal.Methods 2017,9, 3099-3104; Nosrati H. et al, Bioorg. Chemistry 2018, 76, 501-509; Xu S. et al., ACS Applied Materials and Interfaces 2020, 12, 51, 56701–56711).
Prompted by this reviewer comment, we added the following text and further citations in Introduction, lines 66-77, page 2: “BSA in conjunction with SPIONs are exploited for two reasons: (i) achieving a better in vivo biocompatibility (e.g., [25–28]) and (ii) prolonging the blood circulation lifetime of SPIONs, representing MRI nanoprobes (e.g., [29–33]). Both properties are superior in SPIONs in comparison to, for instance, Gd (III) species which are exhaustively reported in literature, even in combination with AuNSs (e.g., [15,16]). Since Gd (III) species are toxic and represent potential risk to environment and human health [34], we rather decided to exploit SPIONs as MRI contrast agents in our nanocomposites.”
It is claimed that the designed nanocomposite system AuBSA-Fe is applicable for magnetic resonance and luminescent imaging. However, no fluorescence imaging data is provided.
Our response: Prompted by this comment, we softened the title (line 2, page 1) as well as the text in line 453 (page 13) by the word “potentially”, we also stressed the green synthesis within the title (line 3, page 1), changed “can” to “could” on page 12 (line 421), and partially reformulated the first sentence in Conclusions (page 14).
To be honest, there is our ongoing research dealing with several cell lines (non-cancerous vs. cancerous) and optimization of parameters of nanocomposites application. It will be a biochemically and biologically oriented study, thus, it would not fit the journal Nanomaterials. In the present manuscript, we rather focused our attention on the synthetic approach, which is completely new, performed detailed characterization of the novel bimetallic nanocomposite and tested its potential applicability as a magneto-luminescent dual probe.
As for luminescence imaging, it is very important to investigate the conditions for excitation and emission maxima and determine luminescent quantum yield of the nanocomposites, which we did. Subsequently, the fluorescence imaging parameters can be setup accordingly. However, the reviewer is true that there are no data about fluorescence imaging included. Therefore, we made the above-mentioned changes in the text of our manuscript.
Authors need to perform MTT assay before coming to the conclusion that AuBSA-Fe nanocomposite is non-toxic.
Our response: We intended to perform MTT assay, which is the most frequently used assay in cytotoxicity evaluation. However, MTT assay (and other assays based on formazan) can overestimate the viability of cells when serum albumin present, leading thus to false-positive results as warned in (Funk D. et al., Serum albumin leads to false-positive results in the XTT and the MTT assay, Biotechiques 2007, 43, 178-182). Furthermore, in (Neufeld B. H. et al., Small molecule interferences in resazurin and mtt-based metabolic assays in the absence of cells, Analytical Chemistry 2018, 90, 11, 6867–6876), it is shown that significant conversion of the assay reagents (dyes) occurs in the presence of many small molecules mimicking thus cellular activity in systems with no cells present at all. Thus, great care have to be taken in viability tests evaluation.
Generally, it is better to use fluorescence, rather than absorbance as for spectroscopic technique, due to the higher sensitivity of the former. Therefore, we rather choose Alamar blue assay (exploitation of resazurin and fluorescence measurements) for our cytotoxicity determination in the present study, similarly as we did in our previous paper concerning AuNCs-BSA biocompatibility assessment (Andrýsková et al., Nanomaterials 2020). The results of cell viability for AuNCs-BSA-SPIONs are now included in our manuscript on pages 13-14, Table 3. It should be, however, pointed out that the results may be still false-negative because resazurin is able to interact with serum albumin, especially at elevated protein concentrations as discussed in (Goegan P. et al., Effects of serum protein and colloid on alamar blue assay in cell cultures,Toxic. In Vitro 1995, 9, 257-266). The results of cell viability can be thus underestimated and need to be verified further by another assay. Indeed, it is the reason why we did not include it in our submitted version of the manuscript. However, prompted by several reviewers, we decided to show and comment it in our revised manuscript. Therefore, it can be found out in the revised version on pages 13-14, lines 454-484.
Significant increment in the particle size of AuBSA from 266 to 351 nm is observed when AuBSA-Fe is formed. This strongly indicates the occurrence of aggregation. It is surprising why the fluorescence quantum yield is reduced only marginally from 6.4 to 6.2%, since Fe is a good quencher?
Our response: We reconsidered the presentation of our DLS data based on this comment. As recommended also by another reviewer, we rather present Z-average values (meaning average hydrodynamic diameter of particles) and polydispersity (PDI) of AuBSA and AuBSA-Fe nanocomposite solutions. Therefore, new Table 1 is included and discussed within the main text (page 8, lines 319-321). Since our previous Figure 3 (showing average PSD based on intensity) would be possibly misleading for readers, we moved it into Supporting Information (currently presented as Figure SI-2) and added detailed explanation of its presentation. In fact, there is no obvious aggregation of AuBSA-Fe in comparison to AuBSA; both samples are stable, they differ just by the Z-average values and PDI (increased in AuBSA-Fe) which is related to the presence of SPIONs.
In order to address possible luminescence quenching, it depends if one consider Fe (II) and Fe (III) interacting with BSA and/or SPIONs being created on BSA simultaneously with luminescent AuNCs. We agree with the reviewer that Fe (II) and Fe (III) are quenchers of Trp molecules within BSA (Xu X. et al., Oxygen-dependent oxidation of Fe (II) to Fe (III) and interaction of Fe (III) with bovine serum albumin, leading to a hysteretic effect on the fluorescence of bovine serum albumin, J. Fluorescence 2008, 18, 193-201). Moreover, Fe (III) ions were selectively detected by quenching the luminescence of AuNCs in several studies (Zhang J. et al, J. Mater. Chem. B 2017, 5, 5608, doi: 10.1039/c7tb00966f; Zhang Y. et al, Journal of Nanoscience and Nanotechnology 2016, 16, 12179-12186, doi: 10.1166/jnn.2016.13772; Ungor D. et al, Colloids and Surfaces B: Biointerfaces 2017, 155, 135-141, doi: 10.1016/j.colsurfb.2017.04.013), but BSA was not the template of these AuNCs. Furthermore, there are no overt studies about SPIONs being quenchers of luminescent AuNCs while generated simultaneously (because it is one novelty of this manuscript). Therefore, a few sentences were added in the beginning of Section 3.1. – page 6, lines 263-274: „There might be concerns about luminescence quenching induced by iron cations since luminescent AuNCs have been used as sensors of Fe (III) in solution ([41–43]. However, in the cited studies, BSA is not used as the template for luminescent AuNCs formation. Moreover, there is a big difference if (i) Fe cations are present in the course of luminescent AuNCs formation within BSA (this study) vs. (ii) they are added to the well-formed luminescent AuNCs ([41–43].“
The manuscript need to be thoroughly checked by English native speaker as many sentences are grammatically incorrect and require appropriate modification (some are highlighted below).
In Abstract: “one-pot sequential green synthesis is developed”……. “By alkalization in the right moment”
On Page 1, Line 41: “with the first type of the approaches to”
On Page 5, Line 219-220: “The intensity of luminescence only very slightly decreased”
Our response: The sentences highlighted by the reviewer were modified (in Abstract lines 18-20 and 23-24; in Introduction lines 37, 44-45; page 7, line 281). Furthermore, we asked an English linguist to check our revised version of the manuscript.
Submission Date
17 February 2023
Date of this review
28 Feb 2023 10:48:52
Faithfully yours,
Assoc. Prof. Karolína Šišková
On behalf of the Authors
Dept. of Experimental Physics, Faculty of Science, Palacký University Olomouc, CZ karolina.siskova@upol.cz

Reviewer 4 Report
A simple one-pot synthesis is developed for the production of nanocomposites containing Au(III) and Fe(II)/Fe(III) ions. It is shown that they have relaxivity values close to those of commercial contrast agents. I think this research is worth pursuing further in order to get nanocomposites at least equally valuable as the existing commercial contrast agents.
Author Response
We thank to all five reviewers for their valuable comments, suggestions, and questions. It greatly improved our manuscript that is now submitted in its revised version. Please, find below our responses made step-by-step to each reviewer comment.
All changes are also marked in the revised version of our manuscript by using track & trace mode and all revisions shown. The numbering of pages and lines in our responses to reviewers is related to the revised version of our manuscript with all revisions shown.
REviewer 4
Open Review
( ) I would not like to sign my review report
(x) I would like to sign my review report
English language and style
( ) English very difficult to understand/incomprehensible
( ) Extensive editing of English language and style required
( ) Moderate English changes required
(x) English language and style are fine/minor spell check required
( ) I don't feel qualified to judge about the English language and style
|
Yes |
Can be improved |
Must be improved |
Not applicable |
|
|
Does the introduction provide sufficient background and include all relevant references? |
(x) |
( ) |
( ) |
( ) |
|
Are all the cited references relevant to the research? |
(x) |
( ) |
( ) |
( ) |
|
Is the research design appropriate? |
(x) |
( ) |
( ) |
( ) |
|
Are the methods adequately described? |
(x) |
( ) |
( ) |
( ) |
|
Are the results clearly presented? |
(x) |
( ) |
( ) |
( ) |
|
Are the conclusions supported by the results? |
(x) |
( ) |
( ) |
( ) |
Comments and Suggestions for Authors
A simple one-pot synthesis is developed for the production of nanocomposites containing Au(III) and Fe(II)/Fe(III) ions. It is shown that they have relaxivity values close to those of commercial contrast agents. I think this research is worth pursuing further in order to get nanocomposites at least equally valuable as the existing commercial contrast agents.
Our response: We thank to the reviewer for the suggestion to continue on this research. Currently we are optimizing the synthesis further and soon will see the results of subsequent tests of MRI contrast.
Submission Date
17 February 2023
Date of this review
27 Feb 2023 20:58:40
Faithfully yours,
Assoc. Prof. Karolína Šišková
On behalf of the Authors
Dept. of Experimental Physics, Faculty of Science, Palacký University Olomouc, CZ karolina.siskova@upol.cz

Reviewer 5 Report
In this work magneto-luminescent bimetallic nanocomposites applicable as dual imaging probes are described. Here one-pot simultaneous bio-mineralization method of gold and iron ions in the presence of BSA under alkaline medium was developed to create new magneto-luminescent probes (AuBSA-Fe). These AuBSA-Fe probes are based on non-toxic BSA-embedded luminescent AuNCs which are generated together with SPIONs simply by alkalization of the reaction mixture. The work is of interest because the great benefit of AuBSA-Fe probes, serving as MR contrasts, lies in their simultaneous luminescent feature. Taking into account the mentioned below notes, I think that the article looks like a short communication and may be published after minor revision.
Notes:
1. There are some printing mistakes That should be checked and corrected. For example, line 59, “in vivo” should be written by italic.
2. I think that DLS data in the form of a table should be presented in the text of the manuscript. In particular, the Z- Average and polydispersity indexes for studied systems should be presented in this table.
3. The general scheme of preparing of bimodal AuBSA-Fe probes should be presented in the manuscript for visual clarity.
4. Promising application of new obtained results should be added in Conclusions.
Author Response
We thank to all five reviewers for their valuable comments, suggestions, and questions. It greatly improved our manuscript that is now submitted in its revised version. Please, find below our responses made step-by-step to each reviewer comment.
All changes are also marked in the revised version of our manuscript by using track & trace mode and all revisions shown. The numbering of pages and lines in our responses to reviewers is related to the revised version of our manuscript with all revisions shown.
Reviewer 5
open Review
( ) I would not like to sign my review report
(x) I would like to sign my review report
English language and style
( ) English very difficult to understand/incomprehensible
( ) Extensive editing of English language and style required
( ) Moderate English changes required
(x) English language and style are fine/minor spell check required
( ) I don't feel qualified to judge about the English language and style
Comments and Suggestions for Authors
In this work magneto-luminescent bimetallic nanocomposites applicable as dual imaging probes are described. Here one-pot simultaneous bio-mineralization method of gold and iron ions in the presence of BSA under alkaline medium was developed to create new magneto-luminescent probes (AuBSA-Fe). These AuBSA-Fe probes are based on non-toxic BSA-embedded luminescent AuNCs which are generated together with SPIONs simply by alkalization of the reaction mixture. The work is of interest because the great benefit of AuBSA-Fe probes, serving as MR contrasts, lies in their simultaneous luminescent feature. Taking into account the mentioned below notes, I think that the article looks like a short communication and may be published after minor revision.
Notes:
- There are some printing mistakes That should be checked and corrected. For example, line 59, “in vivo” should be written by italic.
Our response: We checked and corrected such stupid mistakes. We apologize for them, they happened due to formatting into journal template.
- I think that DLS data in the form of a table should be presented in the text of the manuscript. In particular, the Z- Average and polydispersity indexes for studied systems should be presented in this table.
Our response: We reconsidered the presentation of our DLS data based on this comment. As recommended also by other reviewers, we rather present Z-average values (meaning average hydrodynamic diameter of particles) and polydispersity (PDI) of AuBSA and AuBSA-Fe nanocomposite solutions. Therefore, new Table 1 is included and discussed within the main text (page 8, lines 319-321). Since our previous Figure 3 (showing average PSD based on intensity) would be possibly misleading for readers, we moved it into Supporting Information (currently presented as Figure SI-2) and added detailed discussion and explanation of its presentation.
- The general scheme of preparing of bimodal AuBSA-Fe probes should be presented in the manuscript for visual clarity.
Our response: We generated a comparative scheme of AuBSA and AuBSA-Fe nanocomposites syntheses. It is presented as new Figure 1 in the revised version of our manuscript, the numbers of the other figures are modified accordingly.
- Promising application of new obtained results should be added in Conclusions.
Our response: Based on this comment, we added the final two sentences of Conclusions (page 14, lines 496-500)
.
Submission Date
17 February 2023
Date of this review
27 Feb 2023 13:13:28
Faithfully yours,
Assoc. Prof. Karolína Šišková
On behalf of the Authors
Dept. of Experimental Physics, Faculty of Science, Palacký University Olomouc, CZ karolina.siskova@upol.cz

Round 2
Reviewer 2 Report
The author response all my concerns!
Author Response
Olomouc, 8th March 2023
Dear reviewers,
Dear Dr. James Chow and editorial board,
We definitely thank to the reviewers for their valuable comments and time spent by reviewing our manuscript. All five reviewers helped to improve the manuscript. Since there are no further questions, there are two last changes in our revised manuscript only:
- The title has been changed according to the suggestion of reviewer 3. It now reads: Facile One-Pot Green Synthesis of Magneto-Luminescent Bimetallic Nanocomposites with Potential as Dual Imaging Agent
- In acknowledgment, we thank to all five reviewers.
Faithfully yours, Assoc. Prof. Karolína Šišková
On behalf of the Authors
Dept. of Experimental Physics, Faculty of Science, Palacký University Olomouc, CZ karolina.siskova@upol.cz
Reviewer 3 Report
The authors have responded to all my previous queries. Since, most of the concerns are addressed, I recommend acceptance of the manuscript. Besides, I am suggesting a more suitable title for the manuscript as follows-
Facile One-Pot Green Synthesis of Magneto-Luminescent Bimetallic Nanocomposites with Potential as Dual Imaging Agent
Author Response

(The authors gave the same response as above.)
